# Synthesizing Programmatic Policy for Domain Generalization

## Abstract

Deep reinforcement learning has effectively addressed numerous complex control tasks. However, when the environment undergoes changes, such as increasing the number of discs from three to four in the 'Tower of Hanoi', learned policies often struggle with generalization. We propose an algorithm for learning programmatic policies capable of capturing environment variations. In doing so, these policies gain the capability to generalize to instances where certain aspects of the domain exhibit variations—a property we term domain generalization. We design a Domain Specific Language to construct the structure of the policy. Through sampling tasks from a task distribution, we can train the policy with a meta-learning algorithm. Furthermore, our approach incorporates Recurrent Neural Network (RNN) into the structure of the programmatic policy to enhance agent-environment interactions. Experiment results demonstrate the efficiency of our approach across three environments with domain generalization. In addition, the learned policy shows its ability to generalize to tasks under different variations of environments.

## 1 Introduction

Deep reinforcement learning has achieved significant breakthroughs across many tasks. Nonetheless, the capacity to generalize across diverse environments remains a challenge, even for state-of-the-art deep reinforcement learning algorithms (Packer et al., 2018). Agents often become excessively specialized in the training environments, which hinders their ability to generalize to variations within the same environment. More specifically, consider a 'Tower of Hanoi' task, characterized by three pillars and a collection of discs of varying sizes. The primary objective is to transfer all the discs from the source pillar to the target pillar while adhering to specific constraints. The variations of the task occur where the number of the discs changes, such as a control policy learned from the game of 'Tower of Hanoi' with three discs. In case where the policy is evaluated in the 'Tower of Hanoi' with four discs, it struggles to attain satisfactory performance. Building on the concept of domain randomization proposed by Tobin et al. (2017), we term this property as domain generalization. Our aim is to train a policy capable of generalizing to some variations of the environments.

However, the neural networks based policies frequently encounter difficulties when confronting these challenges. Our approach involves synthesizing a programmatic policy as a means to address this challenge. Such a policy consists of a set of conditional statements and controllers acts like a vanilla control policy, producing an action while input an observation. In contrast to a neural network-based policy, a programmatic policy, such as a recursive program used to solve tasks like the 'Tower of Hanoi', can capture tasks of interest, e.g., different patterns could be learned to tackle the domain generalization. Cai et al. (2017); Liu et al. (2018) introduced neural architectures for program semantics learning, however, not considering control problems. Imitation learning is employed to learn programmatic policies (Verma et al., 2018; 2019), which necessitates user demonstrations. Qiu & Zhu (2021) introduced a program architecture search method for generating programmatic policies without the need for a pre-trained oracle. Nevertheless, the existing approach exhibits limited domain generalization when evaluated.

To tackle the challenge, we propose a method to synthesize a programmatic policy for generalizing across different variations of a environment. Building upon the foundation laid by Qiu & Zhu (2021), the programmatic policy adheres to a domain-specific language structure, and the search space for program architecture can be continuously relaxed, allowing the utilization of gradient-

based reinforcement learning algorithms. Our method leverages domain randomization (Tobin et al., 2017; Peng et al., 2018), which samples tasks from a task distribution by randomizing specific domain aspects for each instance. Consequently, we update the programmatic policy's parameters using a meta-learning approach. Furthermore, we incorporate RNN blocks into the structure of the programmatic policy drawing inspiration from the algorithm presented by Duan et al. (2016). The hidden states of the RNN cells are retained across different episodes throughout the agent-environment interactions. The benefit of the mechanism lies in enabling the agent to remember past knowledge and apply it to new tasks.

We benchmark our method against the state-of-art reinforcement learning methods in three environments respectively. The results demonstrate that the programmatic policy trained via our method exhibits domain generalization across different environment variations. It also performs well in the more complex variations that have not been encountered before. Additionally, utilization of RNN in programmatic policy significantly improves the performance.

Our method draws inspiration from a real-world scenario involving real robot tasks. We operate under the assumption that the values of the aspects within an environment are unlikely to exhibit large variability. For instance, consider the 'Tower of Hanoi' environment, where complexity escalates exponentially as the number of plates increases. Completing such a task with thousands of operations becomes nearly impossible, resulting from the risk of operational failure due to real-world environmental influences. More specifically, in the practical scenario, a range of 3 to 8 discs suffices. Therefore, within this paper, the the values of the aspects we consider are all adhere to reasonable range. In other words, the number of actions necessary for task completion will remain relatively stable.

## 2 METHOD OVERVIEW

### 2.1 PROBLEM FORMALIZATION

We consider a task distribution denoted as $p(\mathcal{T}_H)$, representing an environment encompassing $H$ variations. An agent with programmatic policy $\mathcal{P}_{E,\theta}$ is trained to generalize across variations of the environment, where the $\theta$ presents the parameter of the policy. The parameter $E$ denotes the program architecture which can be defined by Domain Specific Language (DSL). Formally, we can model each task $\tau_i \in \mathcal{T}_H$ for $i \in H$ as a Markov Decision Process defined by a tuple $\{S_i, A_i, T_i, R_i\}$ where $S_i$ and $A_i$ denote the environmental observation and action spaces, respectively. Furthermore, $T_i : S_i \times A_i \times S_i \to [0,1]$ represents state transition probabilities, and $R_i : S_i \times A_i \to \mathbb{R}$ quantifies the corresponding rewards when transitioning between states. The objective is to train a policy $\mathcal{P}_{E,\theta}(a_t \mid s_t)$, receiving state $s_t \in S$ and produces an action $a_t \in S$, that can generalize to $\mathcal{T}_H$. The parameter of the policy $\widehat{\theta}_i$ is estimated by maximizing the cumulative discounted reward $E_{s_0,a_0,s_1\cdots\sim\mathcal{P}_{E,\theta}}[\sum_0^\infty \gamma^t \cdot R_i(s_t, a_t)]$ where $\gamma \in (0,1]$. Subsequently, we update the policy $\mathcal{P}_{E,\theta}$.

### 2.2 PROGRAM STRUCTURE

A programmatic policy processes an environmental state as input and computes an action to be executed by the agent. Drawing inspiration from the program architecture search framework (Qiu & Zhu, 2021), we can infer the structure of a programmatic policy denoted as $\mathcal{P}_{E,\theta}$ by DSL.

$$E ::= C \mid if\ B\ then\ C\ else\ E$$

$$B ::= RNN\_Cell(\mathcal{X}) \geq 0$$

$$C ::= \theta_c + \theta \cdot \mathcal{X}$$

Figure 1: DSL for meta programmatic policy

The DSL is represented in Backus-Naur form (Winskel, 1993). A context-free grammar, depicted in Figure 1, is crafted to define the programs to be learned. The non-terminals $E$ and $B$ are non-terminals and evaluated as action values and boolean. $\mathcal{X} \in \mathbb{R}^m$, with $X$ repre-

senting the input variable of the policy and $m$ denoting its dimension. This DSL allows the derivation of programs. As an example, we can deduce a program as $if\ B_1\ then\ C_1\ else\ E_1$ to $if\ B_1\ then\ C_1\ else\ (if\ B_2\ then\ C_2\ else\ E_2)$. The semantics of the program, such as $if\ B_1\ then\ C_1\ else\ E_1$, are computed by a function denoted as $[\![if\ B_1\ then\ C_1\ else\ E_1]\!](x)$, where the variable $x$ serves as input to the **if-else-then** program. yielding a real-valued vector as output. $C ::= \theta_c + \theta \cdot \mathcal{X}$, yielding a real-valued vector as output, is an affine transformation, where $\theta \in \mathbb{R}^{n \cdot |\mathcal{X}|}$ and $n$ represents the dimension of the action spaces.

To ensure differentiability, the program derived from the DSL can be interpreted as a numerical approximation:

$$[\![if\ B\ then\ C\ else\ E]\!](s) = \sigma([\![B]\!](s)) \cdot [\![C]\!](s) + (1 - \sigma([\![B]\!](s))) \cdot [\![E]\!](s)$$

where $\sigma$ represents the sigmoid function. By utilizing the sigmoid function, the **if-else-then** program is transformed into a differentiable expression with binary branch selection. The output of the sigmoid function represents the probability of selecting a particular branch.

## 2.3 TRAINING OF PROGRAMMATIC POLICY

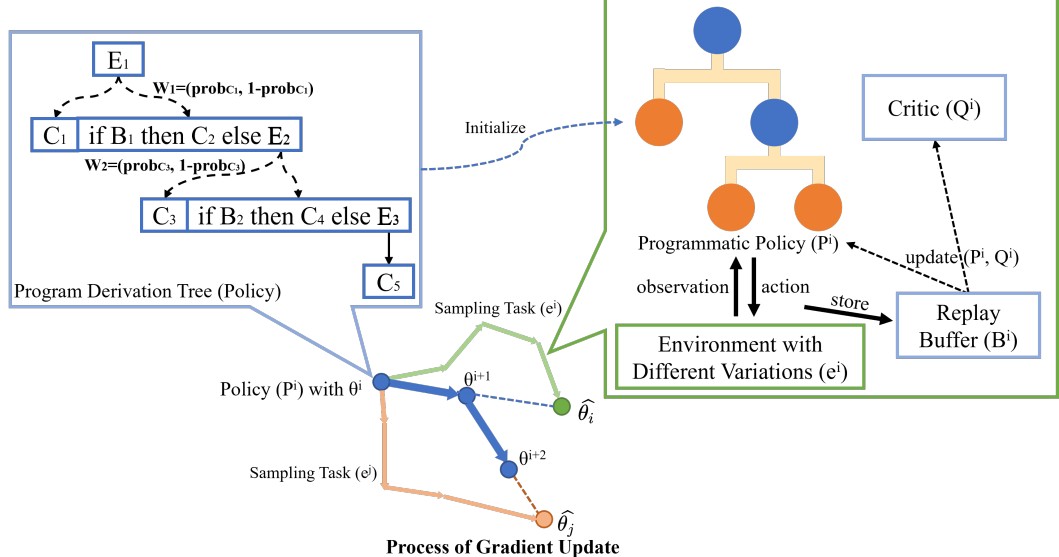

Figure 2: Illustration of training programmatic policy

The left part of Figure 2 illustrates a program derivation tree with a depth of three. According to the DSL expression, we can systematically expand a program into a program derivation tree. The program derivation tree represents all possible program derivations within a specified depth limit for program abstract syntax trees. From the Figure 2, $\mathcal{W}$ is a vector containing the knowledge of the selection of each layer's structure. Each digit in $\mathcal{W}$ models a binary selection. For instance, $prob_{C_1}$ represents the probability of expanding $E_1$ into $C_1$ and $1 - prob_{C_1}$ signifies the probability of expanding it into $if\ B_1\ then\ C_2\ else\ E_2$. The value of $prob_{C_1}$ is calculated using a Softmax function. Thus, the parameter $\theta$ in the programmatic policy $\mathcal{P}_{E,\theta}$ is derived from both the program structure parameter $\mathcal{W}$ and the numerical approximation parameter for **if-else-then** $\varphi$. The parameters $\theta$ can be viewed as a combination of $\mathcal{W}$ and $\varphi$. As $E$ is a constant, we focus on optimizing the $\theta$ parameter for policy $\mathcal{P}_{\theta(\mathcal{W},\varphi)}$.

In this paper, we consider Proximal Policy Optimization (Schulman et al., 2017) as the foundational reinforcement learning algorithm to train the policy as depicted in the right part of Figure 2. The actor $P^i$ represents the programmatic policy we defined, while the critic $Q^i$ is actually a neural network. We utilize a ReplayBuffer $B^i$ for storing trajectories during training.

We adopt Reptile (Nichol & Schulman, 2018) as the learning algorithm for gradient update, which is shown in the bottom part of the Figure 2. It operates by performing a stochastic gradient descent on

---

**Algorithm 1:** Algorithm for learning programmatic policy within domain randomization

---

**Input:** Distribution over tasks $p(\mathcal{T}_H)$, Learning rate $\alpha$, Meta Learning rate $\beta$, DSL $E$, Depth $d$
**Output:** Trained policy $\mathcal{P}_\theta$

---

1   Derive Programmatic Policy $\mathcal{P}_{\theta(\mathcal{W},\varphi)}$ via $(E, d)$
2   Initialize $\theta(\mathcal{W}, \varphi)$ randomly
3   **while** *not done* **do**
4      Sample batch of tasks $\mathcal{T}_i \sim p(\mathcal{T}_H)$
5      **foreach** *task in* $\mathcal{T}_i$ **do**
6         Sample Trajectories $D_i$ by $\mathcal{P}_{\theta(\mathcal{W},\varphi)}$
7         Store in ReplayBuffer $B_i$
8         Estimate $\widehat{\theta}_i$ with learning rate $\alpha$
9      **end**
10      Update $\theta_{i+1}(\mathcal{W}_{i+1}, \varphi_{i+1}) \leftarrow \theta_i(\mathcal{W}_i, \varphi_i) + \beta[\theta_i(\mathcal{W}_i, \varphi_i) - \widehat{\theta}_i(\widehat{\mathcal{W}}_i, \widehat{\varphi}_i)]$
11   **end**
12   Extract $\mathcal{P}_{\theta(\varphi)}$ by fixing an optimal $\mathcal{W}$
13   **while** *not done* **do**
14      Sample batch of tasks $\mathcal{T}_j \sim p(\mathcal{T}_H)$
15      **foreach** *task in* $\mathcal{T}_j$ **do**
16         Sample Trajectories $D_j$ by $\mathcal{P}_{\theta(\varphi)}$
17         Store in ReplayBuffer $B_j$
18         Estimate $\widehat{\theta}_j$ with learning rate $\alpha$
19      **end**
20      Update $\theta_{j+1} \leftarrow \theta_j(\varphi_j) + \beta[\theta_i(\varphi_j) - \widehat{\theta}_j(\widehat{\varphi}_j)]$
21   **end**

---

the sampled tasks and updating the initial parameters toward achieving the final learned parameters specific to the given task. The algorithm solely needs a black-box optimizer such as SGD or Adam and offers good computational efficiency and performance.

In Algorithm 1, our algorithm takes as input training hyperparameters, a DSL description, the maximum depth of the program derivation tree denoted as $d$, and a task distribution. In line 1, we automatically deduce a programmatic policy $\mathcal{P}_{\theta(\mathcal{W},\varphi)}$ in the form of a program derivation tree with depths ranging from 1 to $d$ based on the input DSL. In line 2, we initialize the parameters $\theta$ to be learned. Specifically, we initialize $\mathcal{W}$ with a $50\%$ probability for each branch selection. In line 4, we sample batch of tasks, following the domain randomization. In lines 5 to 9, the $\widehat{\theta}_i$ is estimated. The trajectories are obtained by the interactions between the agent and the environment, in which we design a mechanism to help the agent utilize the RNN structure, will be elaborated upon in the following section. Subsequently, the trajectories derived from these interactions are stored in a ReplayBuffer for training. We employ PPO for optimizing the parameters, and we update both $(\mathcal{W}, \varphi)$ sequentially using a bilevel optimization technique. Following an update of the parameters in the inner loop, we obtain a 'fast' gradient update denoted as $\widehat{\theta}_i(\widehat{\mathcal{W}}i, \widehat{\varphi}i)$. This result is subsequently utilized to update the 'slow' gradient $\theta_i(\mathcal{W}i, \varphi i)$ in the outer loop, as indicated in line 10. Following the principles of Reptile, the expression $\theta_i(\mathcal{W}i, \varphi i) - \widehat{\theta}_i(\widehat{\mathcal{W}}i, \widehat{\varphi}i)$ can be considered as a gradient and subsequently utilized in a more advanced optimizer, such as Adam. Following iterative optimization of $\mathcal{W}$ and $\varphi$, they will stabilize after a certain number of iterations. At this point, we can fix the $\mathcal{W}$ parameters to determine the specific structure of the programmatic policy, as detailed in line 12, wherein we select the structure with the maximum likelihood. From line 13 to 20, we use the similar process to optimize the $\mathcal{P}_\theta$.

## 2.4   USING RNN TO IMPROVE AGENT-ENVIRONMENT INTERACTIONS

The process of the agent interacting with the environment, as described in both line 6 and line 16 of Algorithm 1, is visualized in Figure 3. We sample each task from the task distribution and generate multiple episodes through the agent's exploration. During each episode, the agent engages

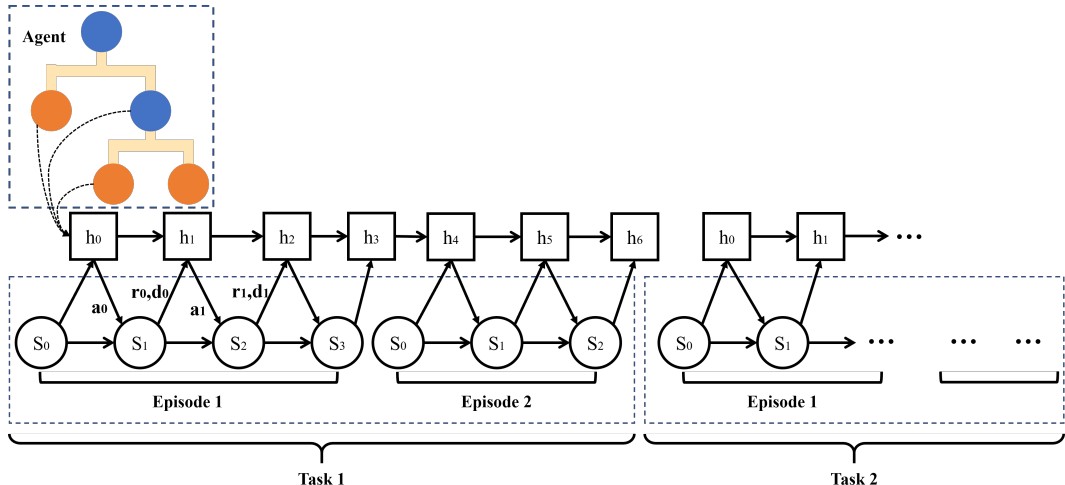

Figure 3: Procedure of agent-environment interaction

with the task environment. Once the agent generates an action $a_t$, the environment provides the corresponding reward $r_t$, advances to the next state $s_{t+1}$, and determines if the episode terminates. This termination status is recorded using the flag $d_t$, which is set to 1 if the episode ends or left at a default value of 0 otherwise. The input is constructed by combining the following elements: the next state $s_{t+1}$, action $a_t$, reward $r_t$, and termination flag $d_t$.

As the task changes and different policies are required for different MDPs, the agent must adjust its actions according to the MDP in which it believes it is currently located. Thus, the agent aggregates all available information on past rewards, actions, and termination flags and continually adapts its policy. To facilitate the agent in learning from prior experiences, the programmatic policy incorporates an RNN structure. It utilizes actions and rewards from preceding time steps as training inputs. According to the DSL 1, certain blocks of the policy comprise recurrent neural network cells. The hidden state $h_t$ is a vector summarized from the programmatic policy. Using the hidden state $h_{t+1}$ and input state $s_{t+1}$ as inputs, the policy generates the subsequent action $a_{t+1}$ and updates the subsequent hidden state $h_{t+2}$. The policy's hidden state is retained across episodes but is not carried over between distinct tasks.

## 3 EXPERIMENTS AND EVALUATION

We design experiments to answer the following questions:

- Can our learned programmatic policy outperform existing algorithms in benchmark tests? Several variations can be derived from these benchmarks. How does our method generalize across them?

- Our approach utilizes meta-learning to train the programmatic policy and integrates RNN blocks into the policy structure. What is the impact of these two ideas on improving generalization?

**Benchmarks**  We assess the effectiveness of our approach using three challenging benchmarks, depicted in Figure 4. These benchmarks—Hanoi, Stacking, and Hiking—are adapted from PDDL-Gym Silver & Chitnis (2020), where both action and state spaces in the simulated environments are discrete. Agents must execute a series of actions to achieve their task objectives.

- **Hanoi**: In this scenario, three adjacent pillars are denoted as A, B, and C, and they hold various-sized disks stacked in a pyramid formation on pillar A. The goal is to methodically transfer all the disks, one at a time, to pillar B. It's crucial to ensure that a larger disk never rests on top of a smaller one within the same pillar. Each action involves relocating a disk

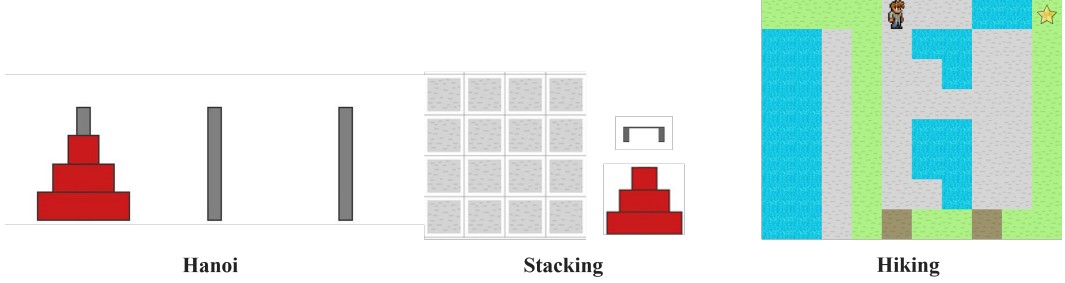

Figure 4: Benchmarks for experiments

from the current pillar to another. The primary variation in this environment is determined by the number of disks involved.

- **Stacking**: On the tabletop, multiple plates of varying sizes are scattered. The task is to systematically pick up these plates one by one using a gripper and assemble them in descending order of size. For simplification, the tabletop is depicted as a grid. The gripper has the capability to move incrementally across the grid or seize a plate. Variations in this environment arise from differing quantities of plates and their respective positions.

- **Hiking**: In the context of this environment, a character embodies the role of the agent within a map, with a star symbolizing the target destination. The agent can traverse pathways, while blue areas denote impassable water obstacles. The overarching goal here is for the character to progress incrementally, collecting all the stars dispersed across the map. Variations in this environment pertain to the quantity and positioning of these stars.

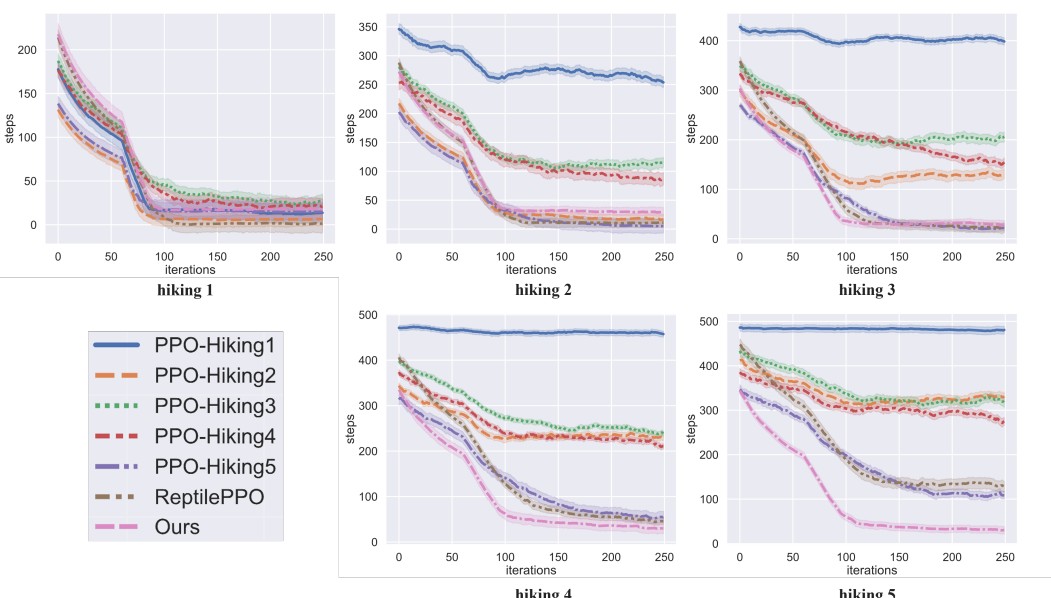

Figure 5: The Performance of PPO, Reptile and our method in Hiking environments with the number of stars ranging from 1 to 5.

**Performance** For the first question, we employ reinforcement learning algorithms: PPO, ReptilePPO, Our method across the three environments. To facilitate the character in collecting all the stars on the map within the hiking environment, we utilize these three methods. The performance in alternative environments is detailed in the supplementary materials. During the training of the ReptilePPO algorithm and our approach, we account for the varying aspects within the hiking envi-

ronment, encompassing the number of stars present (ranging from 1 to 5) as well as their respective locations. Additionally, we train five PPO policies, each tailored to scenarios with a distinct number of stars, ranging from 1 to 5. For instance, `PPO-Hiking1` corresponds to an agent trained using PPO in an environment containing one star. As depicted in Figure 5, each policy is also evaluated in the these environments respectively. The vertical axis signifies the number of steps necessary for the agent to attain its objective, while the horizontal axis denotes the number of iterations involving agent-environment interaction episodes. Performance evaluation of policies is based on the number of steps needed to accomplish a task; a lower step count indicates superior policy performance.

Examination of the figure reveals that policies trained with PPO generally yield commendable performances. Notably, when the map features a modest number of stars, the majority of policies exhibit strong performance. Particularly for tasks involving one star, they almost attain stable convergence. Nonetheless, collecting all the stars within these environments presents a formidable challenge. Policies trained in one-star environments exhibit poor performance when transitioning to environments with two stars. Similarly, in more intricate environments, they struggle to achieve the goal of collecting all the stars. Policies trained in multi-star environments can generally complete tasks across a spectrum of star counts, ranging from 1 to 5. However, the task necessitates hundreds of steps, a condition that is evidently suboptimal.

Policies trained with ReptilePPO and our method exhibit strong performance across environments featuring star counts from 1 to 5. This can be attributed to both policies being trained within a meta learning framework with domain randomization. The agents engage with multiple variations throughout the training process, prompting the agent capture the task of interest.

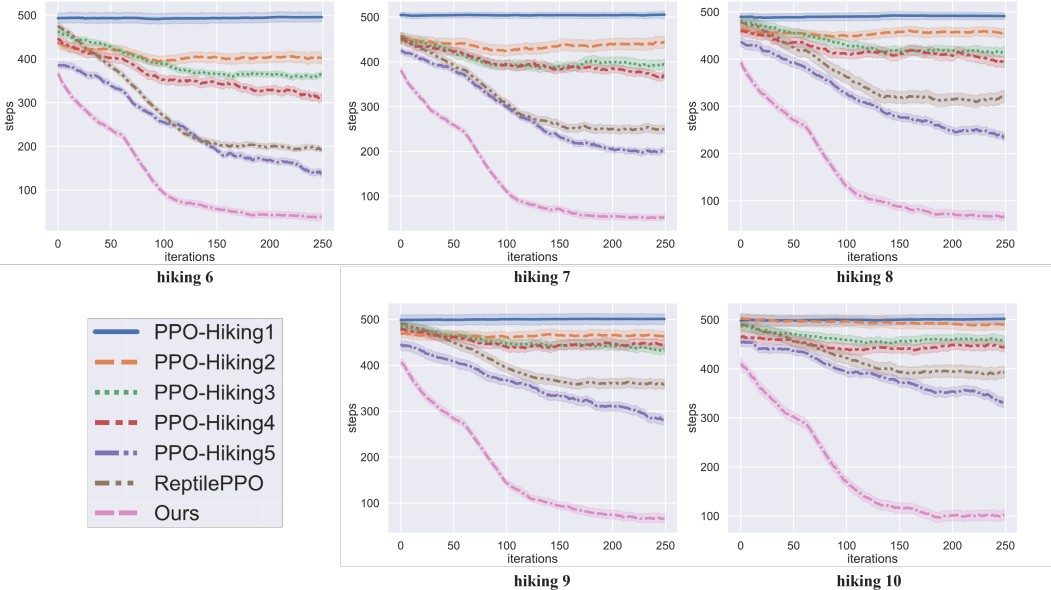

Figure 6: The Performance of PPO, Reptile and our method in Hiking environments with the number of stars ranging from 6 to 10.

Figure 6 illustrates the performance of these policies in environments characterized by greater complexity. In more intricate environments, the performance of the five policies trained with the PPO algorithm deteriorates. The threshold for the number of agent-environment interactions is set at 500. For instance, the ppo-hiking1 policy engages with the environment approximately 500 times, signifying the agent's inability to complete the task within these environments. Similarly, other PPO policies trained in environments featuring a greater number of stars also demand approximately 400 interactions to achieve task completion.

Evidently, policies trained with ReptilePPO outperform those trained with PPO. When confronted with previously unseen scenarios encountered during training, the agent demonstrates a degree of

generalization by completing the task in roughly 200 to 300 steps. Nonetheless, it exhibits sub-par performance in intricate scenarios necessitating the collection of all 10 stars, with an average completion time exceeding 400 steps.

Policies trained through our method consistently exhibit commendable performance, achieving convergence across a broad spectrum of environments variations. Remarkably, even within the most challenging scenario, which involves collecting all 10 stars, agents trained via our method accomplish the task in approximately 100 steps. The performance in these environments with previously unseen variations during training indicates that our method has good generalization when facing unseen scenarios.

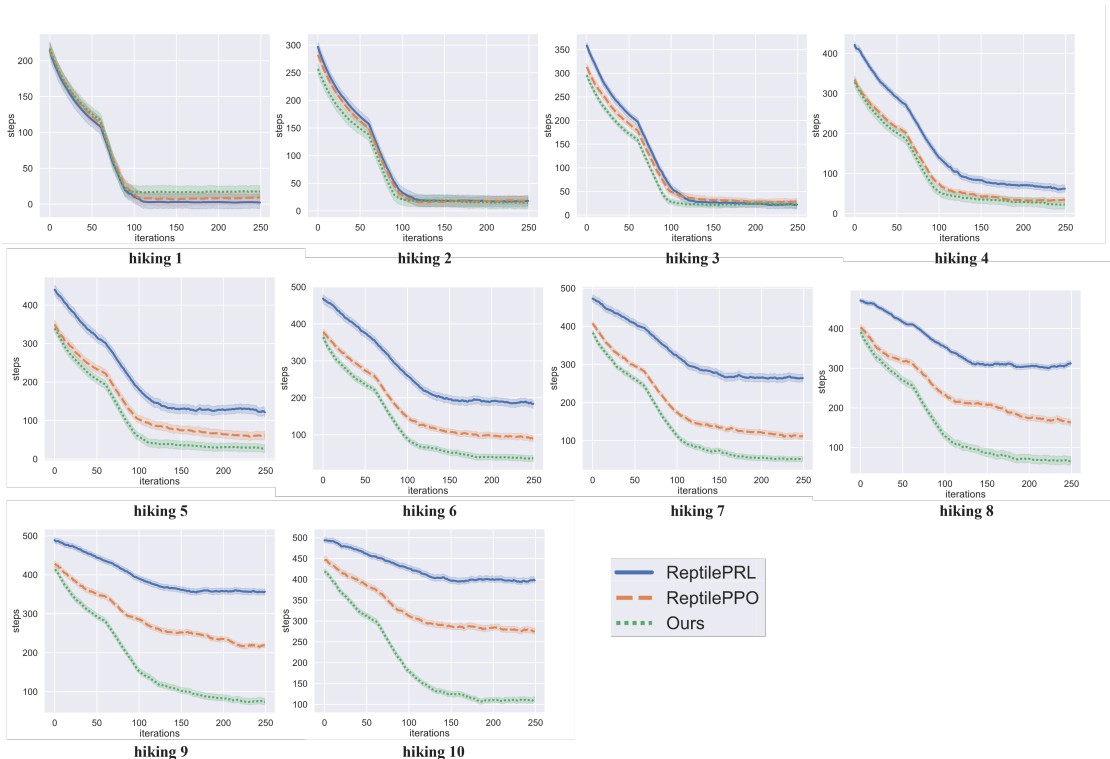

Figure 7: The Performance of ReptilePPO, ReptilePRL and our method in Hiking environments

**Ablation Study**   We investigate the impact of two ideas: the combination of a meta-learning framework with programmatic policy and the utilization of RNN blocks in policy structure construction. We implement ReptilePRL, which trains a programmatic policy without RNN blocks through meta-learning, while the agent-environment interactions adhere to conventional methods.

From Figure 7, the programmatic policy combined with meta-learning can yield relatively better performance than the policy trained by ReptilePPO. In straightforward scenarios, their performances are comparable and have reached an optimal level. However, in more intricate scenarios, particularly those that are unseen, programmatic policies tend to outperform other alternatives. This confirms our initial hypothesis that programmatic policies may be more effective in handling domain generalization. In the comparison between our method and ReptilePRL, both approaches attain commendable performance in simple scenarios. However, as scenario complexity escalates, ReptilePRL struggles to sustain an optimal solution in complex scenarios, whereas our method consistently converges to a satisfactory solution. This is likely due to the fact that when the agent interacts with environments and preserves the RNN's hidden state between each episode, the agent trained by our method effectively leverages prior knowledge and transfers it to new tasks. These results signify that the incorporation of RNN blocks into programmatic policies effectively enhances policy generalization across different variations.

## 4    RELATED WORK

The interaction of an agent and an environment generates data for reinforcement learning. Thus, it is believed that generalization in reinforcement learning is weak, and the model is more prone to overfitting to the current training environment. The two most commonly ways to improve the generalization of reinforcement learning models are regularization and randomization.

Liu et al. (2019); Farebrother et al. (2018) claim that L2 regularization can produce better results than entropy regularization, and L2 regularization can find a good balance point for model's ability and generalization. In robotics, models that perform well in simulators tend to exhibit reduced performance in the real world. Cobbe et al. (2019) introduces CoinRun, an open-source game environment designed to test the generalization performance of deep reinforcement learning algorithms. A regularization parameter is proposed as a positive role to improve the model's generalization. Lu et al. (2020) considers deep reinforcement learning models as two parts: the perception layer and the decision-making layer. An information Bottleneck approach is proposed to constrain the information transmitting, due to the perception layer being more prone to overfitting to the current training environment.

On the other hand, Peng et al. (2018) adds randomization to the simulator during the training phase. Dynamic randomization is adopted to disturb the simulated training environment and to randomize the dynamic parameters of the environment. Akkaya et al. (2019) introduces the Automatic Domain Randomization (ADR) algorithm to address the issue of models trained in simulated environments performing poorly in real environments. Furthermore, Mehta et al. (2020) use Active Domain Randomization to make randomization more efficient by learning the adjustment environment parameters. Tzeng et al. (2020) treats the transition from simulator to reality as a transfer learning problem. The real world robotic controller is learned by the ideas of domain adaptation and paired image alignment. These method use environment randomization for sim2real problem. Packer et al. (2018) claims that environment randomization is the most effective method so far to improve generalization ability based on experiments on several sets of MuJoCo. However, there are potential issues with increasing environment randomization, including: increased complexity of the environment, increased complexity of training and dramatically increased variance.

Most of these methods are not suitable for the domain generalization within the environment with different variations. In our method, we consider employ meta-learning to train a policy. The meta-learning Duan et al. (2016); Finn et al. (2017); Nichol et al. (2018) is employed to allow the agent to quickly learn new tasks based on existing knowledge. Espeholt et al. (2018) proposes a large-scale reinforcement learning training framework, with high performance for multi tasks. Besides, our preference is to use a combination of programmatic policy and meta-learning frameworks for domain generalization. The programmatic policy is mainly inspired by (Qiu & Zhu (2021); Liu et al. (2018)).

## 5    CONCLUSION AND FUTURE WORK

We introduce a framework aimed at learning programmatic policies utilizing a meta-learning algorithm designed to address domain generalization. The policy exhibits a degree of generalization to variations within an environment. To enhance programmatic policy training, we incorporate domain randomization. Additionally, we harness RNN blocks for constructing the programmatic policy, enabling the utilization of previously acquired knowledge from diverse tasks. Experimental results validate the efficacy of our method in acquiring a programmatic policy capable of generalizing across various environmental variations, even extending to previously unseen tasks during training.

Building upon our current method, two avenues for future research emerge. The first involves expanding the applicability of our method to more intricate scenarios, thereby enabling the learned policy to generalize across a broader spectrum of variations. The second direction entails exploring the inclusion of additional grammars into the DSL, resulting in a programmatic policy endowed with a more intricate structure capable of addressing more complex tasks.

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
