# SUPPLEMENTARY MATERIALS

## 1 FORMULATIONS FOR ALGORITHM

This section serves as an introduction for the underlying formulation of the algorithm. Since we use a meta reinforcement learning to train the programmatic policy, the training procedure consists of an outer loop and an inner loop. We use Reptile as the meta reinforcement learning framework to update the parameter in outer loop. For inner loop, we use Proximal Policy Optimization (PPO) as the basic reinforcement learning (RL) algorithm to learn "fast" parameter for outer loop.

In our method, we use PPO-Clip, which utilizes a clipping mechanism in the objective function to remove incentives that may prompt a new policy towards diverging greatly from an existing policy. The PPO-Clip algorithm does not incorporate a KL-divergence term in its objective function. The PPO-Clip objective is:

$$\theta_{k+1} = \arg\max_{\theta} J(\theta)$$

$$= \arg\max_{\theta} \frac{1}{|\mathcal{D}_k| T} \sum_{\tau \in \mathcal{D}_k} \sum_{t=0}^{T} \min \left( \frac{\mathcal{P}_{\theta}(a_t \mid s_t)}{\mathcal{P}_{\theta_k}(a_t \mid s_t)} A^{\mathcal{P}_{\theta_k}}(s_t, a_t), \quad g\left(\epsilon, A^{\mathcal{P}_{\theta_k}}(s_t, a_t)\right) \right)$$

, where

$$g(\epsilon, A) = \begin{cases} (1+\epsilon)A & A \geq 0 \\ (1+\epsilon)A & A < 0 \end{cases} .$$

To illustrate, $\mathcal{D}_k$ represents a set of trajectories by running the policy $\mathcal{P}_{\theta_k}$ in the environment. The $A^{\mathcal{P}_{\theta_k}}$ is the advantage estimation based on the value function and the hyperparameter $\epsilon$ corresponds to how far away the new policy can go from the old while still profiting the objective. The $\theta$ is the parameter for the programmatic policy $\mathcal{P}_{\theta}$, which indeed are two parameters $\theta = (\mathcal{W}, \varphi)$. The $\mathcal{W}$ represents the policy architecture as well as the $\varphi$ is the parameter for policy actions. These two parameters are jointly optimized by PPO. The train process is a bilevel iterative optimization process. At each iteration k of training, we carry out two steps. During the first step, we optimize the $\varphi$ while preserving the frozen architecture parameter weights $\mathcal{W}$:

$$\varphi_{k+1} = \arg\max_{\varphi} J(\mathcal{W}_k, \varphi)$$

Next, during the second step, we optimize the $\mathcal{W}$ by fixing $\varphi$:

$$\mathcal{W}_{k+1} = \arg\max_{\mathcal{W}} J(\mathcal{W}, \varphi_k)$$

The bilevel training steps alternate throughout the training iterations until the reward converges. Then we choose the optimal $\mathcal{W}$ to freezing the architecture of the programmatic policy. Finally, we train the parameters of the selected architecture, continuously until the parameter values learned by means of RL converge to $\widehat{\theta}$.

In the outer loop, the "fast" parameter is used for updating the "slow" parameter. Multiple gradient descents are applied on each task in order to obtain the corresponding parameter value $\widehat{\theta}_i$. Afterwards, the difference vector between the parameters of each task and the main task is calculated as the update direction:

$$\theta_{i+1} \leftarrow \beta(\theta_i - \widehat{\theta}_i)$$

, where $\beta$ is the meta learning rate. By repeating this process iteratively, the global initialized parameters are finally obtained. Intuitively, by using the gradient of a single task's parameter as the rough direction of gradient descent and decreasing the total loss function of the training tasks roughly but steadily, one can often obtain a good initialization parameter.

## 2 IMPLEMENTATION DETAILS

In this section, we present the details of implementation, which cover the input and reward structures of each environment, architecture design and training hyperparameters.

### 2.1 ENVIRONMENT DETAILS

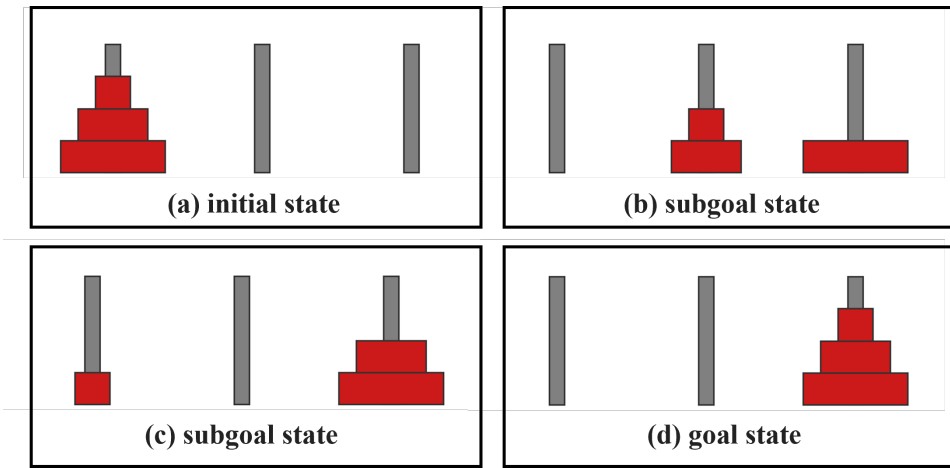

Figure 1: Details for sub-goals in hanoi

**Hanoi**   In the Tower of Hanoi environment, there are three adjacent pillars labeled A, B, C, and several different-sized discs stacked in a pyramid shape from bottom to top on pillar A. The objective is to move all the discs one by one to pillar B, but at no time can a larger disc be placed on top of a smaller one on the same pillar. Each action is to move a disc from the current pillar to another pillar.

In the Tower of Hanoi environment, the aspect of the variations is the number of discs. For different environment variations, the goal of the task is to move all the discs to another pillar. Based on the number of discs on each pillar, as well as the location of the discs, based on their size, we encode the input of this environment into a vector with a dimension of $1 \times 9$. Regardless of the number of discs, we can only set 6 actions, i.e., A to B, A to C, B to A, B to C, C to A, and C to B.

The Tower of Hanoi environment is a discrete environment with low input and output dimensions. However, completing such a task requires the agent to sequentially complete a series of actions, and a small number of mistakes in the action sequence can result in very low rewards for the whole task, or even the inability to complete it. Thus, we set some intermediate rewards for hanoi environment to assist the agent in learning the policy more easily. As shown in the Figure 1, for a Tower of Hanoi environment with three discs, we set (b) and (c) as sub-goals, the environment returns a small reward when the agent accomplishes this for the first time. When the agent moves a disc once, the reward value is -1. When the agent performs an invalid action, such as placing a large disc on top of a small disc, the environment will judge that the action has failed, keep the environment state unchanged, and give a penalty of reward value -2 to the agent. When the agent completes the task, i.e. placing the discs in the state shown in (d), the environment gives a large reward value.

**Stacking**   In the Stacking environment, there are several differently-sized plates on the table, and a gripper that can move above the table. The goal is to collect them one by one using the gripper, putting them together in descending order of size.

In the Stacking environment, the aspects of the environment are the number, size, and location of the plates. To simplify, we use a grid to represent the table. The gripper can move on the grid in up, down, left, and right directions, and can grab a plate. In this environment, we encode the table status, the stacked plates and the location of the gripper as the environment state, with a dimension of $1 \times 35$. The gripper can perform actions of moving up, down, left, and right and selecting a plate to stack and the action is a 5-dimension vector.

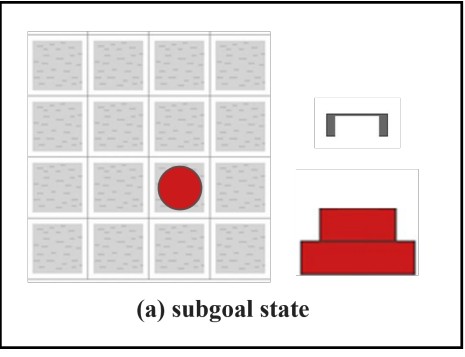 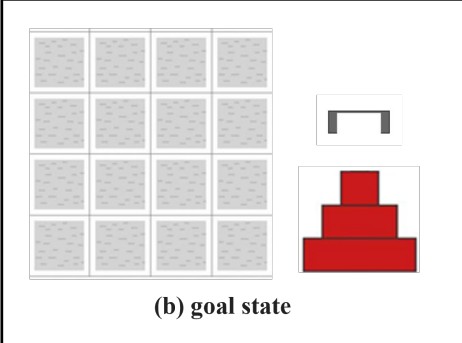

Figure 2: Details for sub-goals in stacking

The Stacking environment is similar to the Tower of Hanoi environment, as the agent needs to control the gripper to select plates in a specific order to complete the task. To reduce the sparseness of rewards, we also set intermediate rewards. For example, as shown in (a) of Figure 2, when the agent first completes stacking two plates, it receives a small reward. Each action taken by the gripper results in a reward value of -1. When the gripper performs an incorrect action, such as moving out of the table or trying to stack plates of wrong size, the environment will judge that the action has failed. Then the environment state keeps unchanged, and give a penalty of -2 to the agent. When the agent completes the task, i.e. stacking the plates in the state shown in (b), the environment gives a positive final reward.

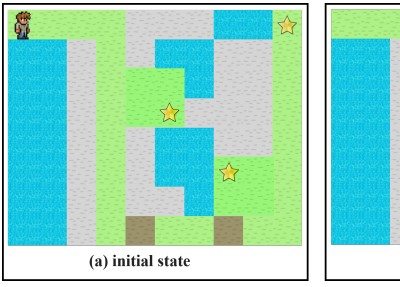 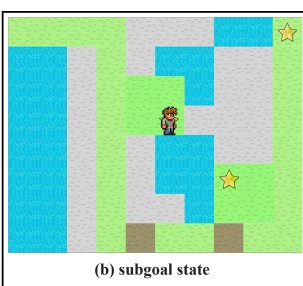 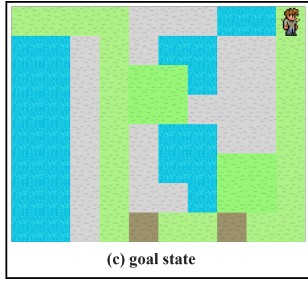

Figure 3: Details for sub-goals in hiking

**Hiking**   In the Hiking environment, the characters on the map represent the agent and the stars on the map are the targets that the agents need to collect. The character can move on the ground, with green and gray blocks indicating walkable ground and blue water blocks representing obstacles that the character cannot pass through. The goal in this environment is to make the character move step by step and collect all the stars on the map.

The aspects of the variations in this environment include the number and location of the stars. The environment is also discretized into a grid. The entire state of the map is encoded as the state of the environment. The character, ground, and water on the map are all marked with different symbols. By flattening the state matrix, we can get a 1x100 vector to represent the state of the environment. The character's actions are up, down, left, and right, so the dimension of the action vector for the environment is $1 \times 4$.

In the Hiking environment, we also set up intermediate rewards, as shown in (b) of Figure 3, where the environment gives the agent a intermediate reward when the character moves to a star and the star disappears from the map. Every time the character take an action in this environment, the environment returns a reward of -1. When the character moves out of the map boundary or into the water, the environment judges the action as a failure and keeps the entire state unchanged, while giving a punishment of reward value -2 to the agent. The final state is shown as in (c).

## 2.2 Hyperparameters

The actor and critic of PPO are both 3-layer linear networks. The input layer is used to process the input of environment state. The middle layer is with a hidden size of 64. The dimension of the output layer is determined by the action space of the environment. Following hyperparameters are used to train PPO algorithm.

- max train steps $2e6$.
- Replay buffer of size 2048.
- Mini-batch size 64.
- Discount factor 0.99.
- Adam optimizer; actor learning rate $3e-4$; critic learning rate $3e-4$.
- clip parameter $\epsilon$ 0.2.
- Entropy coefficient 0.01.
- Max gradient norm 0.5.
- KL-Divergence limit 0.03.

Following hyperparameters are used to train ReptilePPO algorithm.

- max train steps 5e6.
- meta iterations 250.
- inner iterations 1.
- max train steps 2e6.
- Replay buffer of size 2048.
- Mini-batch size 64.
- Discount factor 0.99.
- Adam optimizer; actor learning rate 2e-2; critic learning rate 2e-2.
- Adam optimizer; actor learning rate 1e-5; critic learning rate 1e-5.
- clip parameter $\epsilon$ 0.2.
- Entropy coefficient 0.01.
- Max gradient norm 0.5.
- KL-Divergence limit 0.03.

The components of the programmatic policy are all linear cells. Following hyperparameters are used to train ReptilePRL algorithm.

- depth 6.
- max train steps 5e6.
- meta iterations 250.
- inner iterations 1.
- max train steps 5e6.
- Replay buffer of size 2048.
- Mini-batch size 64.
- Discount factor 0.99.
- Adam optimizer; actor learning rate 2e-2; critic learning rate 2e-2.
- Adam optimizer; actor learning rate 1e-4; critic learning rate 1e-4.
- clip parameter $\epsilon$ 0.2.
- Entropy coefficient 0.01.

- Max gradient norm 0.5.
- KL-Divergence limit 0.03.

In our method, compared with ReptilePRL algorithm, some of the components are GRU layers. And in the updating process, the state of the RNN will be preserved across different episodes. Besides, due to the use of RNN, we have slightly changed the structure of the replay buffer. Following hyperparameters are used to train our method.

- depth 6.
- max train steps 5e6.
- meta iterations 250.
- inner iterations 1.
- max train steps 2e6.
- Replay buffer of size 500.
- Batch size 16
- Mini-batch size 2.
- Discount factor 0.99.
- Adam optimizer; actor learning rate 2e-2; critic learning rate 2e-2.
- Adam optimizer; actor learning rate 4e-4; critic learning rate 4e-4.
- clip parameter $\epsilon$ 0.2.
- Entropy coefficient 0.01.
- Max gradient norm 0.5.
- KL-Divergence limit 0.03.

## 3 ADDITIONAL EXPERIMENTS RESULTS

This section is dedicated to presenting supplementary experimental results and analyses. We apply PPO trained within different variations, ReptilePPO, ReptilePRL and our method in the Tower of Hanoi and the stacking environment respectively.

The performance of these algorithms are shown in Figure 4. We train four PPO policies within the number of plates ranging from 1 to 4 respectively, e.g., `PPO-Stacking1` means the agent is train by using PPO in the environment with one plate. The ReptilePPO, ReptilePRL and our method are also performed in the Stacking environment with the number of plates ranging 1 to 4. And each policy is evaluated in 8 environments with the number of plates ranging from 1 to 8.

From the figure, it can be seen that `PPO-Stacking1`, `PPO-Stacking2`, and `PPO-Stacking3` can basically complete the task, but they are clearly not optimal. Additionally, in more complex scenarios, they do not explore effectively and optimize parameters. For the policy `ReptilePPO`, it can converge to the optimal policy in simple scenarios, but it cannot learn impactful policy in complex environments, such as stacking6, stacking7, and stacking8. For the methods that use the programmatic policy structure, such as ReptilePRL and our method, they do not only perform well in simple scenarios but can also converge to the optimal policy in complex or unseen environments. This indicates that learning RL policies in the form of program structures is useful for improving generalization. However, there is not much difference in performance between our method and ReptilePRL, indicating that the use of RNN structures in this type of environment is not effective.

We also apply PPO, ReptilePPO, ReptilePRL and our method in the Tower of Hanoi environments with the number of discs ranging from 1 to 4. The performance of each policy is evaluated in 7 environments with the number of discs ranging from 1 to 7, as shown in Figure 5.

Through the figure, we can see that the `PPO-Hanoi1` can hardly finish this task in complex environments. This is understandable because as the number of discs increases in the Hanoi Tower environment, the number of steps in the actions will increase exponentially. As the complexity of the environment increases, most policies trained with PPO perform poorly. However, the policies

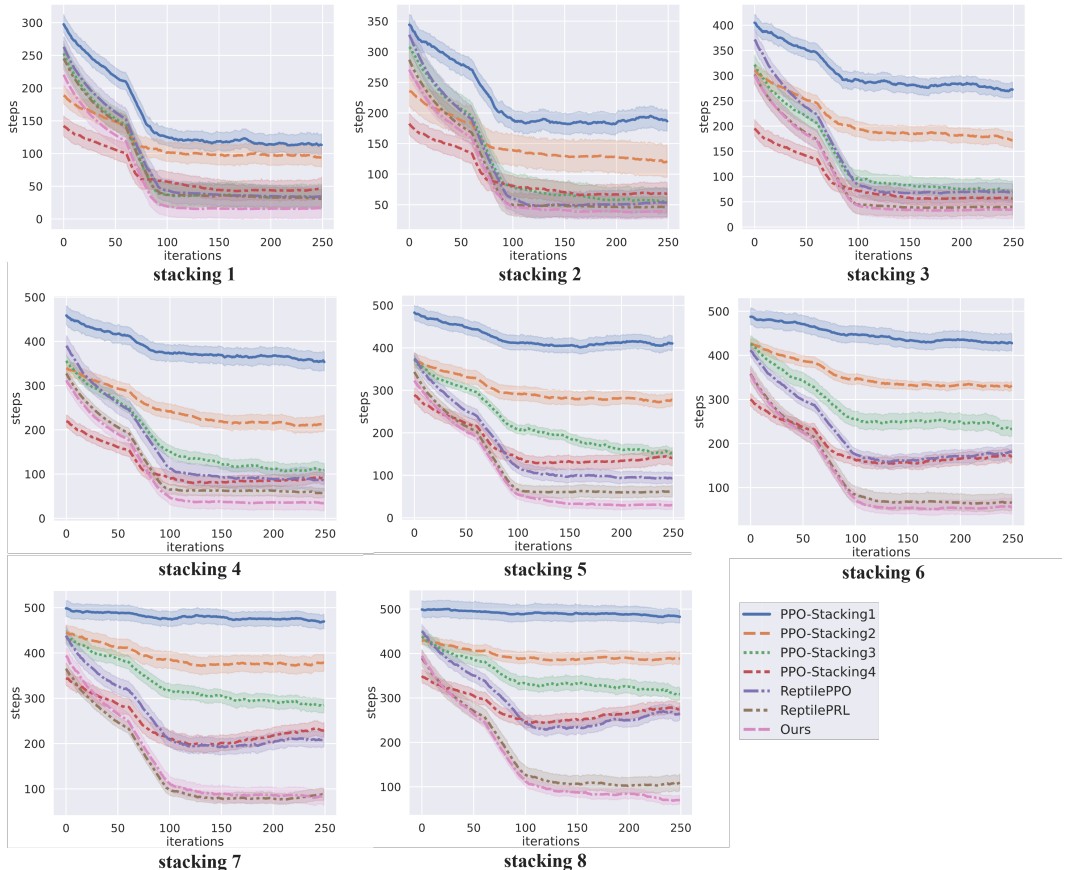

Figure 4: The Performance of PPO, ReptilePPO, ReptilePRL and our method in the Stacking environments with the number of plates ranging from 1 to 8.

trained with meta reinforcement learning can have better performance in complex environments. The performances of `ReptilePPO` and `ReptilePRL` are similar. Our method also performs well in complex scenarios, with an optimal number of steps of 127 for hanoi7, and our method almost reaches the optimum. Compared with ReptilePRL, this demonstrates the effectiveness of the RNN structure we used.

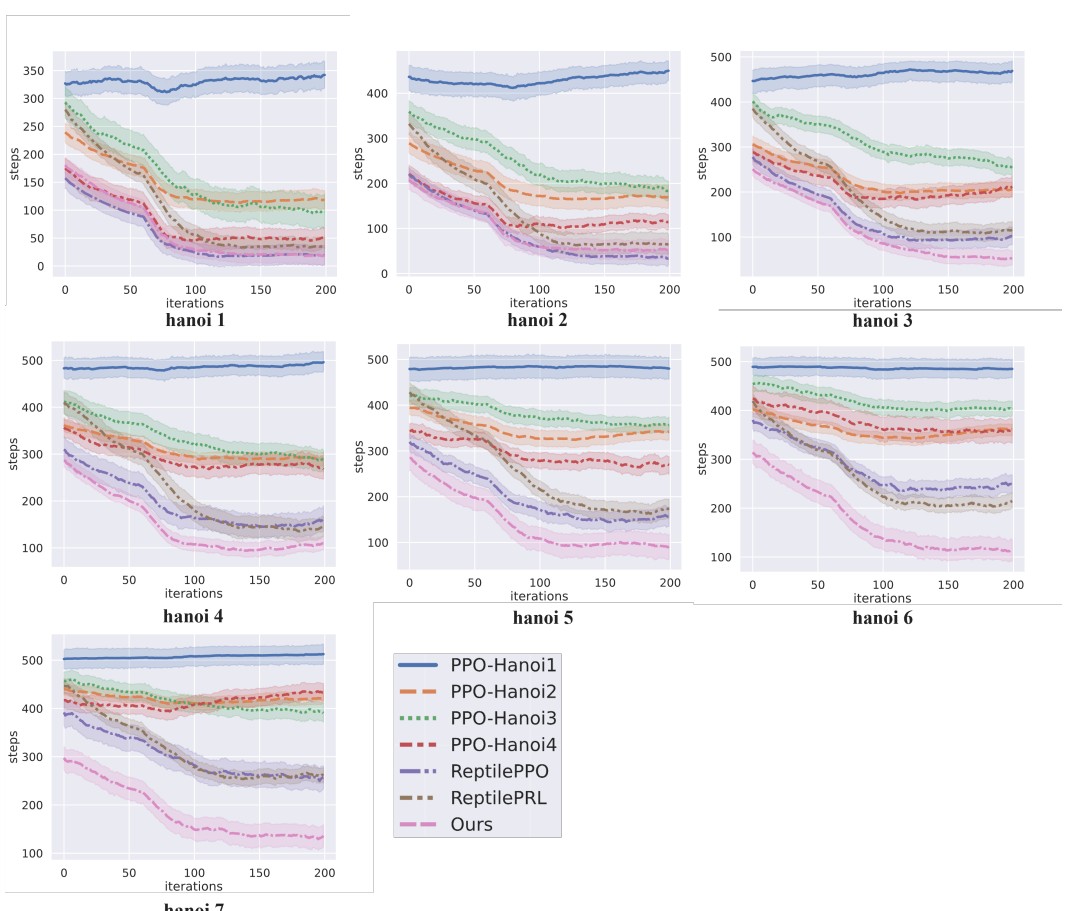

Figure 5: The Performance of PPO, ReptilePPO, ReptilePRL and our method in the Tower of Hanoi environments with the number of discs ranging from 1 to 7.