# OpenReview forum: "Synthesizing Programmatic Policy for Domain Generalization"
_ICLR.cc/2024/Conference — Submitted to ICLR 2024_

### Official Review · Reviewer_F5nm · 2023-10-29

**Soundness:** 3 good
**Presentation:** 3 good
**Contribution:** 3 good
**Rating:** 5
**Confidence:** 4

**Summary:**

This paper addresses the problem of domain generalization in reinforcement learning, especially for RL tasks like "Hanoi Tower", "Stacking," and "Hiking" from the PDDL Gym benchmark. To improve domain generalization across different variations of a given task, the paper proposes a meta-learning framework that incorporates RNN blocks to retain information across different episodes.  The experiments show that during fine-tuning on more complex variations of the given task, the proposed framework can converge more quickly and solve the problem with fewer timesteps than other Deep RL baselines. I believe this work provides an interesting finding about improving domain generalization using programmatic policies and meta-learning. I would like to see this paper included in the conference. However, my major concern about the current submission is the lack of comparison with other programmatic RL approaches and the unspecified number of seeds used in the experiment.  I hope the authors will also address my questions, including providing a more comprehensive analysis of the proposed method on different tasks and adding missing relevant works. The rating is subject to change based on the author's response and update for the current submission.

**Strengths:**

**Motivation and intuition**
- The motivation for employing programmatic policies and meta-learning to address domain generalization problems is reasonable and convincing.

**Novelty**
The idea of employing programmatic policies and meta-learning to address domain generalization problems is reasonable and interesting. This paper proposes a practical way to implement this idea.

**Technical contribution**
- Programmatic Policies for improving domain generalization seem effective, especially in comparison with Deep RL approaches.

**Clarity**
- The overall writing is clear. The authors utilize figures well to illustrate the ideas. Figure 2,3 and Algorithm 1 clearly show the details of the proposed framework.

**Ablation study**
- The proposed framework includes the use of programmatic policies and using RNN blocks for meta-learning.
The ablation studies both of these design choices and provides helpful analysis.


**Experimental results**
- The experimental results are presented in Figure 5-7, which are understandable and clear for comparison.

**Reproducibility**
- Given the pseudocode in the main paper and further details about training hyperparameters in the supplementary materials, I believe reproducing the results is possible.

**Weaknesses:**

**Method**
- The design of DSL is relatively simple, which only consists of “if-then-else” patterns. This DSL may not be capable enough to be applied to harder RL tasks. Other common flow control syntaxes like the “For” and “While” loops may be needed to describe more complicated MDP problems.

**Related work**
- There are plenty of prior works not mentioned in the paper that use similar DSL to address motion control and inductive generalization problems in RL. The Lack of discussion and comparison with these programmatic RL approaches makes it hard to assess the contribution of the proposed framework. Here is the line of works that also uses programmatic policies for RL tasks:
- Programmatic Reinforcement Learning without Oracles
- Synthesizing Programmatic Policies that Inductively Generalize
- Hierarchical Programmatic Reinforcement Learning via Learning to Compose Programs
- Discovering symbolic policies with deep reinforcement learning
- Programmatically Interpretable Reinforcement Learning
- Verifiable Reinforcement Learning via Policy Extraction
- Learning to Synthesize Programs as Interpretable and Generalizable Policies
- Differentiable Synthesis of Program Architectures


**Experiment setup**
- The paper only compares the proposed method against Deep RL approaches, which is not sufficient. Since this work is built upon the paper "Programmatic Reinforcement Learning without Oracles", why not take it as a baseline for the comparison? Other programmatic RL papers (e.g., Synthesizing Programmatic Policies that Inductively Generalize) also address problems of domain generalization in RL, and any additional discussion or comparison with these works will make this work more convincing.

- The comparison between the proposed framework and PPO-N does not seem to be fair. The proposed framework is trained across different variations of the same tasks, while PPO-N is only trained on a single version of the given tasks. I know such a comparison is helpful to show the difference in training with/without domain randomization. However, adding another DRL baseline (e.g., PPO-Hiking1-5) that trained with the same setting as the proposed approach would make the experimental result more convincing.

- The proposed method is only evaluated on "Towers of Hanoi", "Stacking" and "Hiking", while the PDDL Gym benchmark has about 20 different tasks. It would be better to see more evaluation of the proposed framework for these different tasks.



**Experiment details**
- No description of the number of random seeds used for the experiment can be found throughout the paper, which makes it hard to assess the robustness and stability of the proposed method. How many random seeds are used to report the experimental results of Figure 5, 6, 7 in the main paper (and Figure 4 and 5 in the supplementary material)?

**Questions:**

As stated above.

---

### Official Review · Reviewer_iY3T · 2023-11-01

**Soundness:** 2 fair
**Presentation:** 2 fair
**Contribution:** 3 good
**Rating:** 5
**Confidence:** 4

**Summary:**

The paper proposes a reinforcement learning approach for generalizing across structured modular variations in properties of the environment. The key idea in the paper is to instantiate the RL policy as a domain specific language (DSL) and train the policy through meta-learning by sampling tasks from a task distribution. Experiments on discrete simulated control tasks that involve hierarchical reasoning demonstrate the effectiveness of the proposed model compared to some prior RL approaches like PPO and variants.

**Strengths:**

- The paper targets an interesting problem of generalizing across different structured variations of the environment. This is an important desiderata in several real-world applications, and a topic of active interest in the community.

- The proposed approach consists of different components, the combination of which is novel to the best of my understanding. The policy is instantiated as a domain specific language (DSL), and is trained with PPO through domain randomization in simulation. The training algorithm employs meta-learning which helps in learning a policy adaptable to environment variations.

- The experimental comparisons are performed across different types of tasks, involving logical reasoning, table top manipulation, and maze navigation, demonstrating wide applicability of the proposed approach.

- The detailed description of the overall algorithm (Algorithm 1) is helpful in understanding the approach, which consists of different components.

**Weaknesses:**

- All the simulated environments involve discrete state and action spaces, whereas the approach in my understanding is not limited to a discrete state-space. It should be made clear why this is the case, and if possible, there should be some comparisons in the continuous control scenario as well.

- The experimental setting is based on a benchmark from 2020 and results are shown for only three tasks. The tasks are overall much simple (for example low dim state/action space) compared to standard manipulation and locomotion tasks in MuJoCo-Gym and DeepMind Control Suite. It is unclear if there is any limitation of the approach that prevents it from scaling to these more richer and complex control tasks.

- The paper needs significant revision to improve writing clarity, remove typos etc. In addition, there are several areas of improvement for higher quality plots (fig 6 and 7, the font sizes in the legend are too big compared to the axes labels) and potentially more expressive figure captions. Figure 1 looks weird because it is an equation block, and so should be included within the text and not pasted as a figure with large font size.

- For the baselines, PPO makes sense because the proposed algorithm is implemented based off of PPO. But there should be some comparisons to other relevant RL algorithms as well, like SAC, and those that involve hierarchical policy learning.

**Questions:**

Refer to the list of weaknesses above. In summary:

- Is there a reason why all the environments are discrete state and action space?

- Is it possible to evaluate on standard MuJoCo/DM Control tasks that the RL community typically evaluates on, and which are more complex/richer compared to the ones considered?

- Is there a reason why there are no other baselines except PPO?

---

### Official Review · Reviewer_Ji45 · 2023-11-01

**Soundness:** 2 fair
**Presentation:** 1 poor
**Contribution:** 1 poor
**Rating:** 3
**Confidence:** 4

**Summary:**

This work tackles the domain generalization problem in reinforcement learning, which aims to achieve the capability to generalize to instances where certain aspects of the domain exhibit variations. The main idea is to design the structure of policy using a simple program-like DSL. The main benefit of the proposed DSL is that the DSL can be approximated into a differentiable form which can be updated using meta-learning from a distribution of tasks. The proposed method is evaluated on the Hiking domain, in which it outperformed the PPO and Reptile algorithm.

**Strengths:**

* The paper tackles an important topic in AI
* The idea of learning differentiable programs with meta-learning is interesting

**Weaknesses:**

* Overall, the paper needs more polishing, especially in method section
 The readers will benefit a lot if the authors can improve the clarity. There are too many inconsistencies, missing definitions, errors, and typos throughout the paper. For an itemized list, please refer to the questions.

* The figures only present the experiment results for Hiking, while the paper consistently mentions other two environments: Hanoi and stacking.

* The experiment section is poorly structured. PPO policies with varying numbers of stars can be presented in a separate plot as a ablation study since they are not trained in the same setting as other compared methods: reptile and ours.

* Poor choice of baselines.
The baselines need to be more carefully chosen. PPO and Reptile are general RL algorithm while the proposed method is very specific family of method that utilize program structure for policy. It would be more convincing to compare against other programmatic policy learning approaches.

* More complex environments
 The considered environments are quite simple, which limits the significance of the work. It would be helpful to consider more complex environments.

* Evaluation protocol is unclear
 Since the main goal is achieving strong generalization ability, it would be helpful to clearly describe how challenging the generalization is in the experiment setting. Currently, it’s not very clear

**Questions:**

* $i\in H$ seems to be typo
* $a_t\in S$ -> $a_t\in A$
* the objective (i.e., cumulative discounted reward) does not contain $i$, while the sentence describes the parameter update for task $i$.
* The sentence starting “The non-terminals E and B are non-terminals…” is very unclear. Are both E and B action values? Are both of them boolean?
* $X$ -> $\mathcal{X}$
* what is “input variable of the policy”? This needs definition. How is it different from state?
* the dimension of $\theta$ seems wrong, since the dimensionality does not match for operation $\theta \cdot \mathcal{X}$.

---

### Meta-Review · Area_Chair_mxha · 2023-12-09

**Metareview:**

This paper proposes to leverage programmatic reinforcement learning to improve domain generalization. The authors propose a novel meta-algorithm for learning programmatic policies that generalize well to changes in the domain, such as increasing task difficulty. They evaluate their approach on a benchmark of tasks.

While the reviewers agree that the problem studied by the authors is interesting, there were significant concerns about the limitations of the experiments. The authors only evaluate on a small benchmark of toy problems. A significantly expanded experimental evaluation would be needed to convincingly demonstrate the effectiveness of the proposed approach. There were also concerns about clarity of the paper, lack of discussion of prior work, and missing baselines. Nevertheless, the approach considered by the authors appears interesting, and I encourage them to address these concerns and resubmit their work.

**Justification For Why Not Higher Score:**

The paper only evaluates on toy benchmarks, making it difficult to assess the scalability of the approach to realistic tasks. Furthermore, the paper is missing significant baselines and comparisons to prior work.

**Justification For Why Not Lower Score:**

N/A

---

### Decision · Program_Chairs · 2024-01-16

Reject